

# Controls of longitudinal variation in $\delta^{13}$C-DIC in rivers: A
# global meta-analysis
**K. A. Roach[1], M. A. Rodríguez[1], Y. Paradis[2], and G. Cabana[1]**
[1] {Université du Québec à Trois-Rivières, Département des sciences de l'environnement, C.P.
500, Trois-Rivières, Québec G9A 5H7, Canada}
[2] {Ministère des Forêts, de la Faune et des Parcs, 880, Chemin Ste-Foy, Québec City, Québec
G1S 4X4, Canada}
Correspondence to: K. A. Roach (roackat@gmail.com)
**Abstract**
We conducted a literature survey to investigate controls and spatial and temporal patterns of
$\delta^{13}$C-DIC and deviations between $\delta^{13}$C-DIC and the $\delta^{13}$C signature of DIC at isotopic
equilibrium with the atmosphere ($\Delta\delta^{13}$C-DIC) in streams and rivers throughout the world. We
used generalized additive mixed models to relate $\Delta\delta^{13}$C-DIC and $\delta^{13}$C-DIC in lotic ecosystems to
ecological variables including elevation, Strahler order, and partial pressure of dissolved $CO_2$
($pCO_2$), and to examine seasonal shifts in $\Delta\delta^{13}$C-DIC and $\delta^{13}$C-DIC over a range in latitude.
Elevation, Strahler order, and DIC concentrations explained a large fraction of the variation in
$\Delta\delta^{13}$C-DIC, and these variables plus pH and $pCO_2$ explained much of the variation in $\delta^{13}$C-DIC.
Seasonal fluctuations in $\delta^{13}$C-DIC were most apparent in rivers located in temperate regions with
seasonal snow cover. Small streams tended to have lower $\delta^{13}$C-DIC values than large rivers.
Overall, our analysis indicates that processes that add $CO_2$ to the water column, including
groundwater inputs, decomposition, and respiration, should have a greater influence on $\delta^{13}$C-DIC



than processes that remove $CO_2$. Both physical (gas exchange with the atmosphere, weathering,
ice cover) and biological (respiration in regions with high $C_4$ grass abundance, photosynthesis by
cyanobacteria) processes appear to control $\delta^{13}C$-DIC in streams and rivers, but the relative
importance of these processes shifts from upstream to downstream.
Key words: biogeochemistry, carbon, generalized additive mixed model, river, stable isotope

## 1 Introduction

Stable isotope analysis of carbon ($\delta^{13}C$) is widely used to investigate biogeochemical
cycling and food web dynamics in aquatic ecosystems. Geological, atmospheric, and biological
sources of dissolved inorganic carbon (DIC) often have unique isotopic signatures (Boutton
1991); consequently, $\delta^{13}C$ is a valuable tracer of the origin of DIC. The $\delta^{13}C$ signature of DIC
($\delta^{13}C$-DIC) in the water column can be altered by the addition of DIC with a distinctive $\delta^{13}C$
signature and by processes that affect the relative abundance of $^{13}C$:$^{12}C$ (fractionation). The
resulting spatial variation in $\delta^{13}C$-DIC yields important information about the carbon cycle. In
addition to being of interest to biogeochemists, large-scale spatial gradients of $\delta^{13}C$ have been
used by ecologists in trophic applications. Agreement in $\delta^{13}C$ between consumer tissues and DIC
along a fluvial isotope gradient has been used to estimate the spatial scale of consumer feeding
movements (Rasmussen et al., 2009; Bertrand et al., 2011). The $\delta^{13}C$ signature of algae often
tracks the longitudinal gradient in $\delta^{13}C$-DIC, whereas $\delta^{13}C$ of terrestrial-based detritus is
relatively constant over space (Gray et al., 2011). Recent stable isotope mixing models use the
upstream-downstream gradient in $\delta^{13}C$-DIC to estimate contributions of algal versus terrestrial
production sources to consumer biomass by assuming that consumer $\delta^{13}C$ is a weighted mixture





of algal and terrestrial signature gradients (Rasmussen, 2010). A better understanding of controls
and spatial and temporal patterns of $\delta^{13}$C-DIC in rivers would facilitate the use of $\delta^{13}$C as a
natural tracer.

Gas exchange with the atmosphere is one mechanism that might strongly influence $\delta^{13}$C-

DIC in rivers. Exchange of $CO_2$ between the water and the atmosphere causes fractionation
between different carbonate species that is dependent on pH and temperature (Zhang et al.,
1995). At isotopic equilibrium with the atmosphere, $CO_{2\ (aq)}$ has a lower $\delta^{13}$C signature relative
to $HCO_3^-$ and $CO_3^{2-}$. In streams with neutral to basic pH (dominant in $HCO_3^-$), this process yields
$DIC_{(aq)}$ with a relatively high $\delta^{13}$C signature. Other physical processes influencing $\delta^{13}$C-DIC in
rivers include carbonate mineral weathering and mixing of different water bodies. Carbonate
weathering, the formation of $HCO_3^-$ via the dissolution of carbonate minerals, produces DIC with
a relatively high $\delta^{13}$C signature, reflecting $\delta^{13}$C of carbonate rocks (Kendall and Doctor, 2003).
Influx of a water body also can influence $\delta^{13}$C-DIC in the recipient system. For example,
groundwater is supersaturated in $CO_{2\ (aq)}$ from soil respiration and decomposition of organic
matter (Wanninkhof et al., 1990), and its input lowers $\delta^{13}$C-DIC.

Biological processes, including heterotrophic respiration, decomposition of organic

matter, and algal primary production also cause changes in $\delta^{13}$C-DIC. Whereas DIC derived
from heterotrophic respiration and decomposition has a low $\delta^{13}$C signature, algae preferentially
use $^{12}$C over $^{13}$C during photosynthesis, yielding DIC with a high $\delta^{13}$C signature (McKenzie,
1985). Other studies have indicated that in productive rivers, high demand for $CO_2$ by algae
promotes invasion of atmospheric $CO_2$ into the water column (Finlay, 2003; Roussel et al.,
2013). Thus, productive rivers tend to have low $pCO_2$ values due to algae uptake and $\delta^{13}$C-DIC
values that are near isotopic equilibrium with the atmosphere ($\delta^{13}$C-$DIC_{equilibrium}$) (Finlay, 2003).



Deviations between $\delta^{13}$C-DIC and the $\delta^{13}$C signature of DIC at isotopic equilibrium with the
atmosphere ($\Delta\delta^{13}$C-DIC) should reveal biological control of $\delta^{13}$C-DIC in streams and rivers.
Sites from rivers with high algal primary production and low $pCO_2$ would be expected to be
closest to $\delta^{13}$C-DIC$_{equilibrium.}$

A number of studies in lotic ecosystems have shown that $\delta^{13}$C-DIC undergoes a

predictable shift from upstream to downstream, with headwaters having lower $\delta^{13}$C-DIC values
than further downstream (Yang et al., 1996; Telmer and Veizer, 1999; Finlay, 2003). In a study
of the Ottawa River Basin in Canada, Telmer and Veizer (1999) found that $\delta^{13}$C-DIC in upland
tributaries dominated by silicate lithology was -16‰, and $\delta^{13}$C-DIC in the lowland main channel
dominated by carbonate lithology was -8‰. Similarly, in a survey of streams and rivers in
Northern California, USA, Finlay (2003) found that $\delta^{13}$C-DIC and $CO_{2\ (aq)}$ increased with
discharge. DIC dynamics in rivers may respond to shifts along the upstream-downstream
gradient in environmental factors. For example, Telmer and Veizer (1999) concluded that soil
respiration and carbonate weathering produced much of the longitudinal variation in $\delta^{13}$C-DIC,
and that $CO_2$ exchange between the water and atmosphere and instream primary production were
relatively unimportant. Finlay (2003) suggested that groundwater inputs, outgassing (evasion) of
excess $CO_2$, and microbial respiration primarily control $\delta^{13}$C-DIC in small streams, whereas
algal photosynthesis and $CO_2$ exchange with the atmosphere become more important in large
rivers. An increase in carbonate weathering, a decline in groundwater inputs, $CO_2$ loss to the
atmosphere, and increasing rates of algal primary production are all associated with a decline in
$CO_{2\ (aq)}$ concentrations with increasing river size (Jones and Mulholland, 1998; Dawson et al.,
2001), making the relative influence of these processes on $\delta^{13}$C-DIC difficult to parse out.

In addition to having high spatial heterogeneity, rivers are temporally dynamic.



Floodplain rivers in particular show strong seasonality resulting from changes in river-floodplain
connectivity. Flood stage has a strong impact on several major processes influencing $\delta^{13}$C-DIC,
including ecosystem metabolism, the fraction of groundwater relative to total discharge volume,
and decomposition of organic matter. Several studies have identified seasonal shifts in $\delta^{13}$C-DIC
that are comparable to longitudinal variation in $\delta^{13}$C-DIC. In lowland reaches of the Okavango
River in Botswana, decomposition of terrestrial organic matter contributes to $\delta^{13}$C-DIC values
that are 4‰ lower during the annual period of floodplain inundation (Akoko et al., 2013). In the
St. Lawrence River in Canada, seasonality in algal primary production and respiration in soils
and groundwater causes $\delta^{13}$C-DIC values to shift from -6.8‰ in spring to -1.0‰ in autumn
(Hélie et al., 2002).

We conducted a literature survey of $\Delta\delta^{13}$C-DIC and $\delta^{13}$C-DIC in lotic ecosystems

throughout the world, with the aim of contrasting the physical and biological processes
controlling $\Delta\delta^{13}$C-DIC and $\delta^{13}$C-DIC in streams and in large rivers. We used generalized
additive mixed models (GAMMs; Wood, 2006) to relate $\Delta\delta^{13}$C-DIC and $\delta^{13}$C-DIC in lotic
ecosystems to ecological variables including elevation, Strahler order, and the partial pressure of
dissolved $CO_2$ (p$CO_2$), and to examine seasonal shifts in $\Delta\delta^{13}$C-DIC and $\delta^{13}$C-DIC over a range
in latitude. We originally expected that if $\delta^{13}$C-DIC was mainly under biotic control, we would
find a negative relationship between p$CO_2$ and $\Delta\delta^{13}$C-DIC.

**2 Methods**
**2.1 Literature survey**

We collated data from the scientific literature using the search engines Google Scholar and

Web of Science and using the search terms "river", "stable isotope", and "dissolved inorganic



carbon". We recorded data directly from tables or interpolated data from figures and maps using
data extraction software (GraphClick v.3.0.2) and Google Earth. For each site we recorded
geographic coordinates, $\delta^{13}$C-DIC, temperature, pH, alkalinity, $pCO_2$, DIC concentrations, and
day-of-the-year of sample collection. If $pCO_2$ was not provided, we calculated it from pH,
alkalinity, and temperature (Stumm and Morgan, 2012). For the studies that only provided the
month or season of sampling, we used the midpoint of month or season as an estimate of day-of-
the-year. We were mainly interested in investigating controls of DIC cycling in the main channel
of rivers and thus eliminated sites from natural lakes. Our search yielded a total of 1,530 $\delta^{13}$C-
DIC values from 26 publications (Appendix 1). Our dataset has a nested structure, with sites (n =
801) nested within rivers (n = 302), which are nested within watersheds (n = 31). Watersheds
were located in Africa (4), Asia (4), Australia (1), Europe (6), North America (7), and South
America (9) (Fig. 1).

The $\delta^{13}$C-DIC signature at isotopic equilibrium with the atmosphere, $\delta^{13}$C-DIC$_{equilibrium}$,

was calculated from Zhang et al. (1995) using the equations $\varepsilon CO_{2(aq)}$ - $CO_{2(gas)}$ = -0.0049 × T -
1.31, $\varepsilon HCO_3^-$ - $CO_{2(gas)}$ = -0.141 × T + 10.78, and $\varepsilon CO_3^-$ - $CO_{2(gas)}$ = -0.052 × T + 7.22, where T is
temperature in °C. We assumed a $\delta^{13}$C signature of -7.8 for atmospheric $CO_2$ (Levin et al. 1987).
The proportion of each DIC species was then weighted using the equation $\delta^{13}C_{DIC} = \sum f_i\varepsilon_i$, where
f is the proportion of each species (i) and $\varepsilon$ is the permil fractionation. We used the proportion of
DIC species estimated from pH and temperature (Stumm and Morgan, 2012) to determine $\delta^{13}$C-
DIC$_{equilibrium}$.

We used a geographic information system to determine elevation (masl), Strahler order,

and channel distance from river mouth for each site. We characterized Strahler order using the
15-sec river network data from HydroSHEDS, a hydrographic database derived from elevation





data (Lehner et al., 2008). The majority of sites were at low elevation (i.e., 82% were at < 500
masl), but the highest-elevation sites (> 3500 masl) were from second or third order streams.
Rivers with Strahler order > 4 accounted for 44% of the sites in our literature survey. In the
northern hemisphere,  < 1% of sites were collected from rivers in polar regions (latitude > 66.5°),
72% were from temperate regions (latitude 23.5° to 66.5°), and 7% were from tropical regions
(latitude 0° to 23.5°) (Fig. 1). In the southern hemisphere, no sites were collected from polar
regions, 5% were from temperate regions, and 16% were from tropical regions.

## 2.2 Generalized additive mixed models

Sites were spatially nested within rivers and watersheds, requiring us to account for intra-

group correlations in our statistical analyses. By adding random effects to the additive predictor,
GAMMs enable modeling of overdispersed and correlated data that frequently arise in spatial
analyses (Wood, 2006). Before constructing GAMMs, we examined correlations among the
explanatory variables using matrix scatterplots and transformed strongly skewed variables to
reduce the influence of extreme values.

We constructed separate GAMMs relating $\Delta\delta^{13}$C-DIC and $\delta^{13}$C-DIC to all the explanatory

variables. We then iteratively removed variables if 95% confidence intervals for the smooth
function included zero throughout the range of measured values, and re-fitted the model until the
zero line was not entirely contained by the confidence interval for all variables. The GAMMs for
$\Delta\delta^{13}$C-DIC and $\delta^{13}$C-DIC both had $\delta^{13}$C-DIC as the dependent variable, but models for $\Delta\delta^{13}$C-
DIC additionally included $\delta^{13}$C-DIC$_{equilibrium}$ as an offset. We allowed for different seasonal
patterns in $\Delta\delta^{13}$C-DIC and $\delta^{13}$C-DIC in the Northern and Southern hemispheres by including the
interaction between day of the year and latitude in the GAMMs. We modeled seasonality and



latitudinal effects as cosinor functions of day-of-year and latitude (Barnett and Dobson, 2010)
comprising sine and cosine terms because measurements on a linear scale do not adequately
represent the circular nature of these variables (e.g., the equal spacing between days-of-year 365,
1, and 2 is not captured by the linear scale).

All GAMMs were fit with an identity link function, a normal distribution for errors, and a

penalized regression spline (Wood, 2006, 2011). We incorporated Strahler order as a simple
linear effect and river and watershed as random effects. Distance from river mouth (river km)
was included as a covariate to account for smaller-scale spatial correlation. We used restricted
maximum likelihood estimation (REML) to fit GAMMs. Potential explanatory variables for
$\Delta\delta^{13}$C-DIC included elevation, Strahler order, $pCO_2$, DIC, and the interaction between latitude
and day of the year. Potential explanatory variables for $\delta^{13}$C-DIC included elevation, Strahler
order, temperature, pH, $pCO_2$, DIC, and the interaction between latitude and day of the year. The
final models were:

$\delta^{13}$C-DIC = $\log_e$(DIC) + $\beta_0$ + $\beta_1$ Strahler order + $f(\log_e$(Elevation + 8)) + $f(\sin(\pi \times$ Day-of-
year/365), $\sin(\pi \times$ Latitude/180)) + $f(\cos(\pi \times$ Day-of-year/365), $\cos(\pi \times$ Latitude/180)) +
$f(\log_e$(Distance from river mouth + 1)) + $f$(Watershed) + $f$(River) + $\varepsilon$

for $\Delta\delta^{13}$C-DIC, and

$\delta^{13}$C-DIC = $\beta_0$ + $\beta_1$ Strahler order + $f$(pH) + $f(\log_e$ ($pCO_2$)) + $f(\log_e$ (Elevation + 8)) + $f(\sin(\pi \times$
Day-of-year/365), $\sin(\pi \times$ Latitude/180)) + $f(\cos(\pi \times$ Day-of-year/365), $\cos(\pi \times$ Latitude/180)) +
$f(\log_e$ (Distance from river mouth + 1)) + $f$(Watershed) + $f$(River) + $\varepsilon$




for $\delta^{13}$C-DIC.

In the models, $\log_e$(DIC) is an offset, $\beta$ are regression coefficients, $f$(.) are smooth

functions, $f$(Watershed) and $f$(River) represent random effects, and $\varepsilon$ is normally distributed
error. We used R software (R Development Core Team, 2015) for all statistical analyses and the
mgcv package to generate GAMMs (Wood 2011).

**3 Results**

In the literature survey, $\delta^{13}$C-DIC ranged from -28.1‰ to 0.7‰. $\delta^{13}$C-DIC$_{equilibrium}$ ranged

from -10.6‰ to 2.9‰, and thus did not capture the full variation in $\delta^{13}$C-DIC (Fig. 2). Many of
the sites from lowland rivers, including the Indus, Negro (Argentina), Okavango, Santa Cruz,
and St. Lawrence rivers, had $\delta^{13}$C-DIC values that were near $\delta^{13}$C-DIC$_{equilibrium}$ (Fig. 2).
However, other sites from lowland rivers, including the Slave and Madeira rivers, had $\delta^{13}$C-DIC
that was considerably lower than $\delta^{13}$C-DIC$_{equilibrium}$.

The GAMM of $\Delta\delta^{13}$C-DIC modeled deviations between $\delta^{13}$C-DIC and $\delta^{13}$C-

DIC$_{equilibrium}$. Explanatory variables retained in this GAMM included elevation, Strahler order,
DIC, and the interaction between latitude and day of the year (Table 1). The final GAMM for
$\Delta\delta^{13}$C-DIC included 889 data points from the literature survey; the model explained a total of
85% of the deviance and showed good agreement between fitted and observed values (Fig. 3).

The GAMM of $\Delta\delta^{13}$C-DIC indicated that sites at the lowest and highest elevations were

closer to $\delta^{13}$C-DIC$_{equilibrium}$ (Fig. 4). $\Delta\delta^{13}$C-DIC values were positively related to river size, as
measured by Strahler order, indicating that streams tended to have $\delta^{13}$C-DIC that was far from





$\delta^{13}$C-DIC$_{equilibrium}$ and large rivers tended to have $\delta^{13}$C-DIC that was near $\delta^{13}$C-DIC$_{equilibrium}$ (Fig.
5). Sites with high DIC concentrations also tended to be near $\delta^{13}$C-DIC$_{equilibrium}$, as shown by
high $\Delta\delta^{13}$C-DIC values (Fig. 4). The relationship between DIC concentration and $\Delta\delta^{13}$C-DIC
was not linear. There was a positive relationship between DIC and $\Delta\delta^{13}$C-DIC at low
concentrations (0.05 to 0.20 mmol/L), and no relationship between DIC and $\Delta\delta^{13}$C-DIC for
concentrations > 0.20 mmol/L. Seasonal variation in $\Delta\delta^{13}$C-DIC was apparent in rivers located in
temperate regions from 40° to 60°N and -60° to -40°S (Fig. 6). In the northern hemisphere from
40° to 60°, $\Delta\delta^{13}$C-DIC values were lowest during the winter and early spring (i.e., from January
to April). In the southern hemisphere from 40 to 60°, $\Delta\delta^{13}$C-DIC values were lowest from
January to August and highest during the spring and early summer (i.e., from September to
December). Tropical rivers tended to have lower $\Delta\delta^{13}$C-DIC values than temperate rivers. The
highest $\Delta\delta^{13}$C-DIC fitted values were for rivers located from 40° to 60°N during September to
December.
Explanatory variables retained in the model for $\delta^{13}$C-DIC included elevation, Strahler
order, pCO$_2$, DIC, pH, and the interaction between latitude and day of the year (Table 2). The
final GAMM for $\delta^{13}$C-DIC included 890 data points from the literature survey; the model
explained a total of 91% of the deviance and showed good agreement between fitted and
observed values  (Fig. 3).
The GAMM of $\delta^{13}$C-DIC showed that sites at the lowest and highest elevations tended to
have high $\delta^{13}$C-DIC values (Fig. 7). $\delta^{13}$C-DIC values were positively related to river size, as
measured by Strahler order (Fig. 5). $\delta^{13}$C-DIC was related negatively to pCO$_2$ and positively to
DIC (Fig. 7).  Again, the relationship between DIC concentrations and $\delta^{13}$C-DIC was nonlinear.
There was a positive relationship between DIC and $\delta^{13}$C-DIC at low concentrations (0.05 to 0.20




mmol/L), and a weakly positive relationship between DIC and $\delta^{13}$C-DIC at higher concentrations
(0.20 to 7.5 mmol/L). The relationship between pH and $\delta^{13}$C-DIC also was nonlinear. Sites with
lowest and highest pH tended to have low $\delta^{13}$C-DIC values (Fig. 7). Seasonal patterns in $\delta^{13}$C-
DIC and $\Delta\delta^{13}$C-DIC were very similar (Fig. 6).

**4 Discussion**

Our main objective was to investigate controls and spatial and temporal patterns of $\delta^{13}$C-

DIC in rivers throughout the world. pCO2 explained a large fraction of the deviance in the
GAMM of $\delta^{13}$C-DIC, but was not retained in the GAMM of $\Delta\delta^{13}$C-DIC, suggesting that algal
primary production has limited control over $\delta^{13}$C-DIC. However, as we will discuss below, our
results provide evidence that other biological processes, including photosynthesis by
cyanobacteria and respiration in regions with high $C_4$ grass abundance are important
determinants of $\delta^{13}$C-DIC in rivers (Fig. 8). Our results revealed changes in the dominant
processes influencing $\delta^{13}$C-DIC from upstream to downstream. Furthermore, we found that $\delta^{13}$C-
DIC changed seasonally in rivers in temperate regions with seasonal snow cover.

$\Delta\delta^{13}$C-DIC values were low ($\delta^{13}$C-DIC was near $\delta^{13}$C-DIC$_{equilibrium}$) in high-elevation

streams and large rivers, indicating that exchange of $CO_2$ between surface water and the
atmosphere occurs via invasion or evasion in these systems. Most studies show that lotic
ecosystems are supersaturated in pCO$_2$ (Richey et al., 2002; Jones et al., 2003; Cole et al., 2007)
and are sources, not sinks, of $CO_2$ to the atmosphere (Mulholland et al., 2001; Battin et al.,
2008). Thus, in most low-order streams, high $CO_{2\ (aq)}$ concentrations originating from inputs of
groundwater and terrestrial soil water from sediments (e.g., Jones and Mulholland, 1998) result
in high $\Delta\delta^{13}$C-DIC values, indicating that $\delta^{13}$C-DIC is far from $\delta^{13}$C-DIC$_{equilibrium}$. In high-





elevation streams, low $\Delta\delta^{13}$C-DIC values have been attributed to the high gradient and low
surface:volume ratio, which increases water turbulence and promotes $CO_2$ outgassing (Rebsdorf
et al., 1991; Dawson et al., 2004). For example, a recent estimate of gas transfer velocities in
streams and rivers throughout the United States found that $CO_2$ outgassing is highest in
headwater streams originating in areas with steep topography (Butman and Raymond, 2011). In
small streams, groundwater consists of a large fraction of stream discharge, but its volume
relative to the total volume of discharge decreases downstream (e.g., Devol et al., 1987; Johnson
et al., 2006). $\Delta\delta^{13}$C-DIC values were likely low in large rivers because groundwater has less of
an influence on water chemistry and because surface water has been exposed to the atmosphere
for a relatively long period of time, promoting evasion of $CO_{2\ (aq)}$. $\Delta\delta^{13}$C-DIC values are
particularly low in lacustrine rivers because the prolonged residence time of water in large lakes
allows $CO_{2\ (aq)}$ to equilibrate with the atmosphere (Yang et al., 1996). $\Delta\delta^{13}$C-DIC values also
were low in rivers with high DIC concentrations (> 0.2 mmol/L). In our dataset, DIC
concentrations were strongly correlated with alkalinity ($r = 0.98$) and $CO_{2\ (aq)}$ concentrations did
not have a strong effect on total DIC. $\Delta\delta^{13}$C-DIC values were probably low in rivers with high
DIC concentrations because the carbon was derived from carbonate dissolution, which has a high
$\delta^{13}$C-DIC signature (Kendall and Doctor, 2003).

$pCO_2$ did not explain a large fraction of the total variation in $\Delta\delta^{13}$C-DIC, providing

evidence that algal primary production does not control exchange of $CO_2$ between surface water
and the atmosphere in most rivers. Calculation of $pCO_2$ from pH, alkalinity, and temperature
results in values that are overestimated in acidic waters rich in dissolved organic carbon (Abril et
al., 2015). In our meta-analysis, $pCO_2$ was calculated from these physicochemical variables in
90% of the studies. Direct measurements of $pCO_2$ in streams and rivers would allow for a more




accurate assessment of the relationship between $pCO_2$ and $\Delta\delta^{13}$C-DIC. Although algal primary
production apparently did not cause invasion of atmospheric $CO_2$, low rates of photosynthesis
may have been sufficient to increase $\delta^{13}$C-DIC values without increasing mixing of DIC between
the water and atmosphere. Some of the unexplained variation in the GAMM analysis of $\delta^{13}$C-
DIC might have been caused by preferential assimilation of $^{12}$C over $^{13}$C by algae.
Our results showed that $pCO_2$ is negatively related to $\delta^{13}$C-DIC. The spatial patterns in
$\delta^{13}$C-DIC indicated by our analysis correspond to patterns in $pCO_2$ in streams and rivers that
have been observed elsewhere. For example, $pCO_2$ in rivers tends to decline downstream
(Raymond et al., 1997; Teodoru et al., 2009; Butman and Raymond, 2011), similar to the
relationship between $\delta^{13}$C-DIC and Strahler order revealed by our analysis. Furthermore, tropical
rivers typically have higher concentrations of $CO_{2\ (aq)}$ than temperate rivers (Aufdenkampe et al.,
2011), similar to our results for latitudinal variation. The cycling of $CO_2$ has a strong influence
on the biogeochemistry of $\delta^{13}$C-DIC; thus, processes that remove or add $CO_2$ to the water
column should also influence $\delta^{13}$C-DIC. Major processes that remove $CO_2$ in lotic systems
include $CO_2$ evasion and algal primary production. In small streams, the major source of $CO_{2\ (aq)}$
is believed to be groundwater and terrestrial soil water, with *in situ* respiration and
decomposition of organic matter becoming more important in large rivers (Johnson et al., 2008;
Aufdenkampe et al., 2011). Most lotic systems are supersaturated in $CO_2$ and typically have low
$\delta^{13}$C values relative to equilibrium with the atmosphere. Therefore, processes that add $CO_2$ to the
water column should generally have a greater influence on $\delta^{13}$C-DIC than processes that remove
$CO_{2.}$
Similar to the spatial patterns in $\Delta\delta^{13}$C-DIC values, high-elevation streams and large
rivers had DIC with a high $\delta^{13}$C signature. The congruent spatial trends between $\Delta\delta^{13}$C-DIC and



$\delta^{13}$C-DIC suggest that gas exchange influenced $\delta^{13}$C-DIC in these systems. Our results also show
that DIC had a high $\delta^{13}$C signature in rivers with high DIC concentrations. Large rivers likely
have high $\delta^{13}$C-DIC values because their waters have higher concentrations of carbonate
minerals, and this higher buffering capacity inhibits decrease in $\delta^{13}$C-DIC. Within a river
network, concentrations of dissolved ions are frequently heterogeneous among headwater
reaches, but tend to average out and increase downstream (Livingstone, 1963). Our results
indicated that sites with low and high pH had DIC with a low $\delta^{13}$C signature. Many of the sites
with the lowest surface water pH (3.8 to 6.5) also tended to have low DIC concentrations (range
= < 0.1 to 2.5 mmol/L, average = 0.6 mmol/L). $\delta^{13}$C-DIC values may have been low in rivers
with acidic surface water because these were blackwater rivers draining forested watersheds
dominated by igneous rock. In addition to having low pH, blackwater rivers have high
concentrations of dissolved organic matter that increase rates of microbial respiration (Meyer
1990), lowering $\delta^{13}$C-DIC values. Sites with high pH and low $\delta^{13}$C-DIC values also had low
$pCO_2$ (sites with pH > 9 had a maximum $pCO_2$ of 115 ppmv). When pH is high and $pCO_2$ is low,
intense photosynthesis by cyanobacteria can increase the rate of $CO_2$ invasion from the
atmosphere, producing fractionation that results in low $\delta^{13}$C-DIC values (Herczog and Fairbanks,
1987). Following the depletion of free $CO_2$ that occurs in highly alkaline waters, cyanobacteria
can float on the water surface and use atmospheric $CO_2$ during carbon fixation (Paerl and Ustach,
1982). Lakes also exhibit a positive trend between pH and $\delta^{13}$C-DIC until pH values of
approximately 8-9, at which point $\delta^{13}$C-DIC decreases (Bade et al., 2004).

Rivers in the northern hemisphere between 40° and 60° exhibited seasonal cycles in

$\Delta\delta^{13}$C-DIC and $\delta^{13}$C-DIC. In these rivers, $\Delta\delta^{13}$C-DIC and $\delta^{13}$C-DIC values were lowest from
January to April, corresponding with the formation of ice cover. Studies of lakes have shown that



duration of ice cover is an important control of seasonal patterns in $pCO_2$ and $\delta^{13}$C-DIC. Ice
insulates the lake from mixing by wind and gas exchange, which causes $pCO_2$ to increase in the
water column and lowers $\delta^{13}$C-DIC values (Striegl et al., 2001; Karlsson et al., 2008). The same
phenomenon has been observed in terrestrial ecosystems, with snow cover increasing
accumulation of $CO_2$ in the soil and lowering $\delta^{13}$C-DIC values in the winter (Aravena et al.,
1992). Ice cover also has been documented to increase $pCO_2$ in the water column of rivers (e.g.,
Raymond et al., 1997), and should be responsible for the seasonal shifts in $\Delta\delta^{13}$C-DIC and $\delta^{13}$C-
DIC values in rivers at high latitudes in the northern hemisphere. Sample size was low in
temperate regions of the southern hemisphere. A greater number of sites sampled from these
rivers may result in a seasonal trend that is similar to the pattern observed in the northern
hemisphere. Our results also revealed that DIC has a lower $\delta^{13}$C signature in many tropical rivers
than in temperate rivers. An analysis of $^{13}$C and $^{14}$C of DIC, dissolved organic carbon, and
multiple particulate organic carbon fractions in Amazonian rivers concluded that high $pCO_2$ was
sustained by *in situ* respiration of terrestrial $C_4$ grasses (Mayorga et al., 2005). Terrestrial $C_4$
grasses decompose more rapidly than terrestrial $C_3$ plants (Wynn and Bird, 2007), and the high
relative abundance of this group of macrophytes in the tropics promotes microbial respiration,
lowering $\delta^{13}$C-DIC values even during low-water periods.
Our analysis indicated a contrast between mechanisms influencing $\delta^{13}$C-DIC in streams
and in floodplain rivers (Fig. 8). Physical and biological processes control $\delta^{13}$C-DIC throughout
the upstream-downstream gradient. However, whereas gas exchange with the atmosphere and
groundwater inputs appear to be dominant controls of $\delta^{13}$C-DIC in small streams, carbonate
weathering and photosynthesis by cyanobacteria are particularly important in higher-order rivers.
To fully evaluate physical and biological controls of $\delta^{13}$C-DIC in streams and rivers,



measurements of rates of algal primary production and respiration, gas exchange between the air
and water, and groundwater inputs would be needed. Our results also have implications for
future studies using $\delta^{13}$C to investigate food web structure in rivers. Studies in temperate regions
with seasonal snow cover should take temporal shifts in $\delta^{13}$C-DIC into account. Failure to
consider temporal changes in $\delta^{13}$C-DIC can bias inferences about consumer-resource dynamics
(Woodland et al., 2012). Finally, food web models that incorporate patterns of spatial variation in
$\delta^{13}$C-DIC (Rasmussen et al., 2009; Rasmussen, 2010) are dependent upon an underlying isotope
gradient. Thus, the applicability of these models in many aquatic ecosystems has been
questioned (Layman et al., 2012). Our GAMM analysis showed a positive relationship between
Strahler order and $\delta^{13}$C-DIC, indicating that headwaters have low $\delta^{13}$C-DIC values relative to
downstream reaches in most river systems. The rivers with the most pronounced upstream-
downstream gradients in $\delta^{13}$C-DIC will be those in which headwaters have high levels of $pCO_2$
(Fig. 8). Incorporation of spatial variation in $\delta^{13}$C-DIC in future lotic food web studies will thus
provide insights about the ecology of aquatic organisms, including movements and assimilation
of organic matter from alternative production sources.










**Appendix A: Sources of data for the literature review**

| Watershed | Location | Citation |
|---|---|---|
| Amazon | South America | Mayorga et al., 2005 |
| Brazos | Texas, USA | Zeng et al., 2011 |
| Chico | Argentina | Brunet et al., 2005 |
| Chubut | Argentina | Brunet et al., 2005 |
| Colorado | Argentina | Brunet et al., 2005 |
| Congo | Africa | Bouillon et al., 2012; Bouillon et al., 2014 |
| Coyle | Argentina | Brunet et al., 2005 |
| Danube | Europe | Kanduč et al., 2007 |
| Deseado | Argentina | Brunet et al., 2005 |
| Ems | Germany | Stögbauer et al., 2008 |
| Fraser | British Columbia, Canada | Cameron et al., 1995; Spence and Telmer, 2005 |
| Gallegos | Argentina | Brunet et al., 2005 |
| Ganges-Brahmaputra | Asia | Galy and France-Lanord, 1999 |
| Han | Korea | Lee et al., 2007 |
| Indus | Asia | Karim and Veizer, 2000 |
| Khwai | Africa | Akoko et al., 2013 |
| Lagan | Ireland | Barth et al., 1998 |
| Mackenzie | Canada | Hitchon and Krouse, 1972; Reeder et al., 1972 |
| Murray | Australia | Cartwright, 2010 |
| Nass | British Columbia, Canada | Spence and Telmer, 2005 |
| Negro | Argentina | Brunet et al., 2005 |
| Okavango | Africa | Akoko et al., 2013 |
| Pearl | Asia | Zhang et al., 2009 |
| Rhine | Europe | Flintrop et al., 1996; Stögbauer et al., 2008 |
| Rhône | Europe | Aucour et al., 1999 |
| Santa Cruz | Argentina | Brunet et al., 2005 |
| Skeena | British Columbia, Canada | Spence and Telmer, 2005 |
| Squamish | British Columbia, Canada | Spence and Telmer, 2005 |
| St. Lawrence | Canada and USA | Yang et al., 1996; Telmer and Veizer, 1999; Hélie et al., 2002 |
| Tana | Kenya | Bouillon et al., 2009; Tamooh et al., 2013 |
| Vistula | Poland | Wachniew, 2006 |








**Acknowledgments**
We are grateful to E. Mayorga for providing data from the Amazon River Basin and to P.
Massicotte for assistance in determining Strahler order. Financial support came from the Plan
d'Action Saint-Laurent (PASL) to YP, Natural Sciences and Engineering Research Council of
Canada (NSERC) Discovery Grants to MAR and GC, and an NSERC Banting Postdoctoral
Fellowship to KAR.

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



Table 1. Quantitative output of the generalized additive mixed models (GAMM) of $\Delta\delta^{13}$C-DIC,
including information on units and model effects. Estimated degrees of freedom = edf. The
GAMM explained a total of 85.2% of the deviance.

| Model term | Unit | Effect | Estimate | Std. Error | edf | Standard deviation | p-value |
|---|---|---|---|---|---|---|---|
| Intercept | | | -10.07 | 0.63 | | | <0.001 |
| Strahler order | | Linear | 0.27 | 0.09 | | | <0.01 |
| Dissolved inorganic carbon (DIC) | mmol/L | Smooth | | | 6.4 | | <0.001 |
| Elevation | masl | Smooth | | | 5.0 | | <0.001 |
| Day-of-year × Latitude | | Smooth | | | 19.9 | | <0.001 |
| Day-of-year × Latitude | | Smooth | | | 14.1 | | <0.001 |
| Distance from river mouth | km | Smooth | | | 7.1 | | <0.001 |
| Watershed | | Random | | | 11.7 | 1.7 | <0.001 |
| River | | Random | | | 97.1 | 1.5 | <0.001 |













Table 2. Quantitative output of the generalized additive mixed models (GAMM) of $\delta^{13}$C-DIC,
including information on units and model effects. Estimated degrees of freedom = edf. The
GAMM explained a total of 90.6% of the deviance.

| Model term | Unit | Effect | Estimate | Std. Error | edf | Standard deviation | p-value |
|---|---|---|---|---|---|---|---|
| Intercept | | | -10.76 | 0.58 | | | <0.001 |
| Strahler order | | Linear | 0.27 | 0.09 | | | <0.01 |
| pH | | Smooth | | | 5.9 | | <0.001 |
| Dissolved inorganic carbon (DIC) | mmol/L | Smooth | | | 7.1 | | <0.001 |
| Partial pressure of dissolved CO$_2$ (pCO$_2$) | ppmv | Smooth | | | 3.8 | | <0.001 |
| Elevation | masl | Smooth | | | 5.1 | | <0.001 |
| Day-of-year × Latitude | | Smooth | | | 21.0 | | <0.001 |
| Day-of-year × Latitude | | Smooth | | | 15.2 | | <0.001 |
| Distance from river mouth | km | Smooth | | | 6.4 | | <0.001 |
| Watershed | | Random | | | 9.9 | 1.4 | <0.001 |
| River | | Random | | | 104.5 | 1.6 | <0.001 |













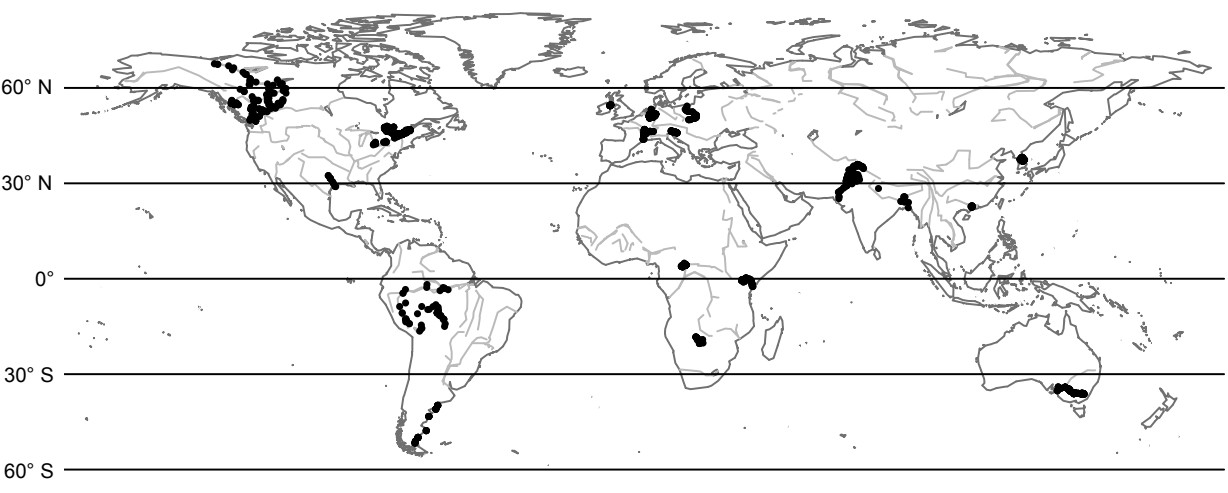


Figure 1. Map of the study sites included in the literature survey. Major world rivers are shown

in light grey.



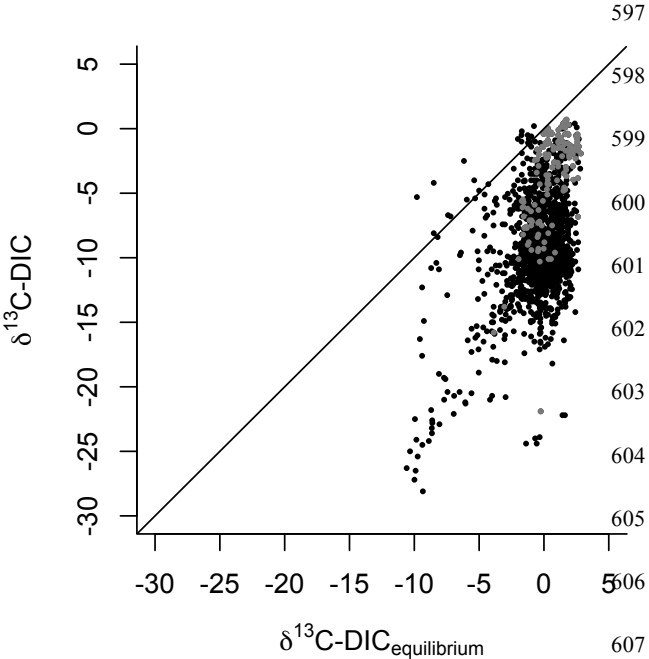

Figure 2. The relationship between $\delta^{13}$C-DIC$_{equilibrium}$ (the $\delta^{13}$C signature of DIC at isotopic

equilibrium with the atmosphere) and $\delta^{13}$C-DIC in rivers. Grey symbols are from rivers with

Strahler order = 8 and black symbols are from rivers with Strahler order < 8. The line indicates

the 1:1 relationship.





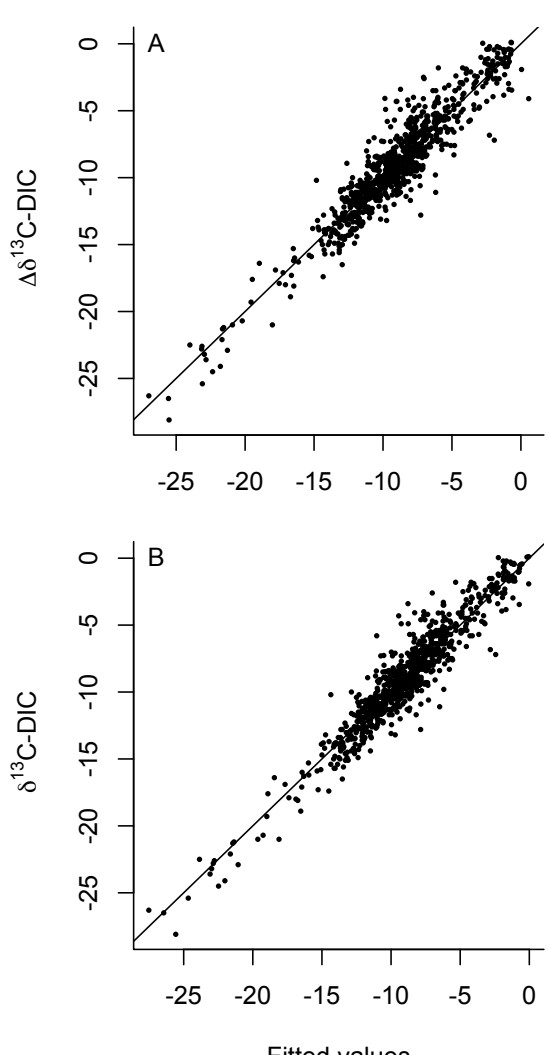


Figure 3. Plots of fitted versus observed values for $\Delta\delta^{13}$C-DIC (A) and $\delta^{13}$C-DIC (B).




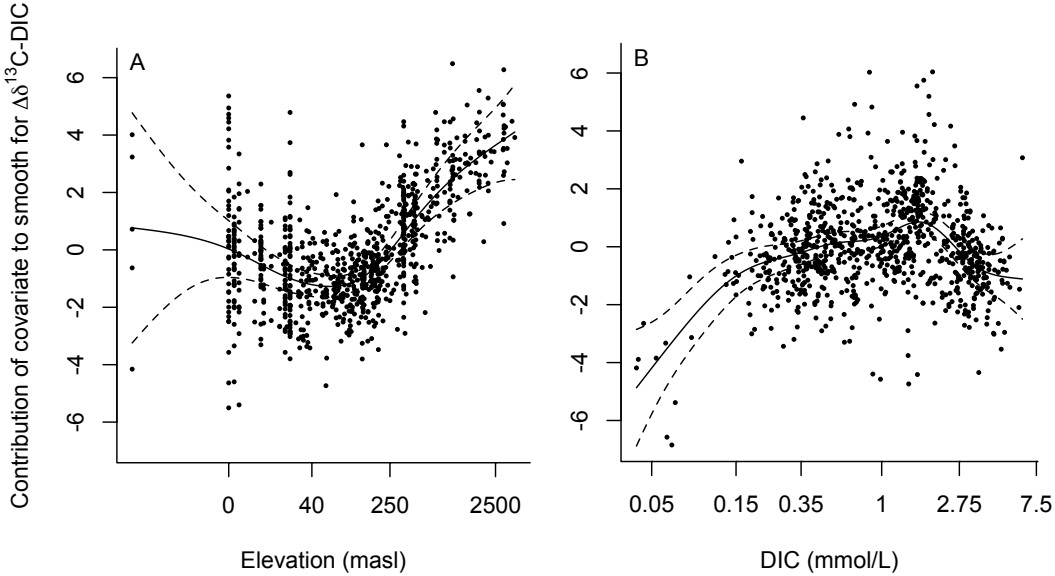



Figure 4. Smooth function (curve) derived from the generalized additive mixed model (GAMM)

fit for relationships between $\Delta\delta^{13}$C-DIC and elevation (meters above sea level, A) and dissolved

inorganic carbon (DIC) concentrations (mmol/L, B). Dashed curves represent 95% confidence

intervals for the smooth. Partial residuals (points) are also shown. Higher $\Delta\delta^{13}$C-DIC values

indicate that $\delta^{13}$C-DIC is closer to isotopic equilibrium with the atmosphere. Note that the

smooth function y-axis is centered to zero mean.










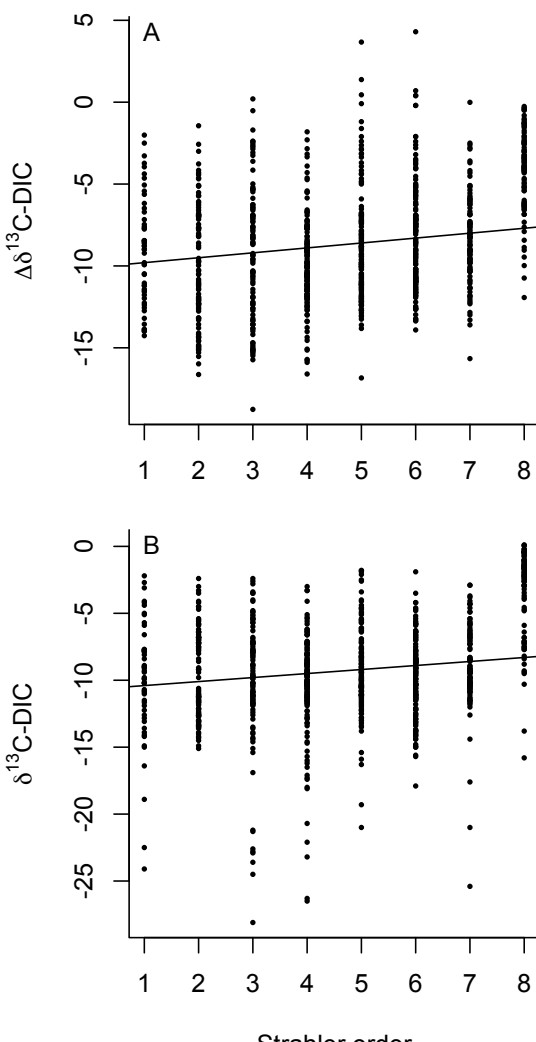

Figure 5. Relationships between Strahler order and $\Delta\delta^{13}$C-DIC (A) and $\delta^{13}$C-DIC (B). The
regression lines determined by the coefficients of the generalized additive mixed models
(GAMMs) are shown.





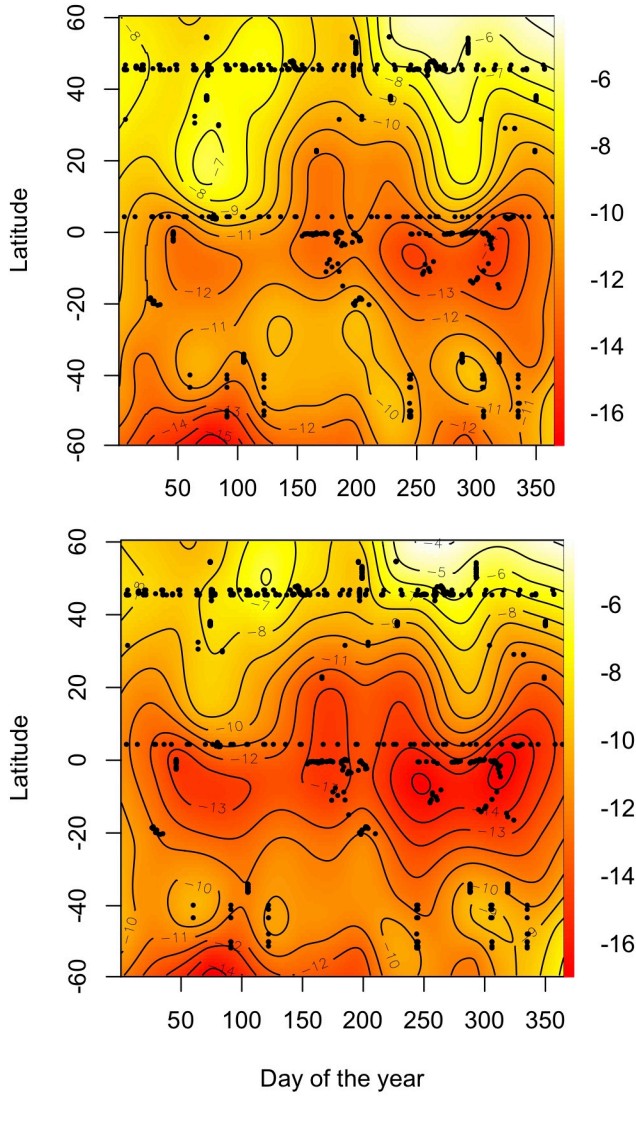


Figure 6. Contour plots showing the seasonal variation in fitted $\Delta\delta^{13}$C-DIC (top) and $\delta^{13}$C-DIC

(bottom) at different latitudes. Lighter colors indicate higher $\Delta\delta^{13}$C-DIC values (values closer to

atmospheric equilibrium) for the top plot and higher $\delta^{13}$C-DIC values for the bottom plot.




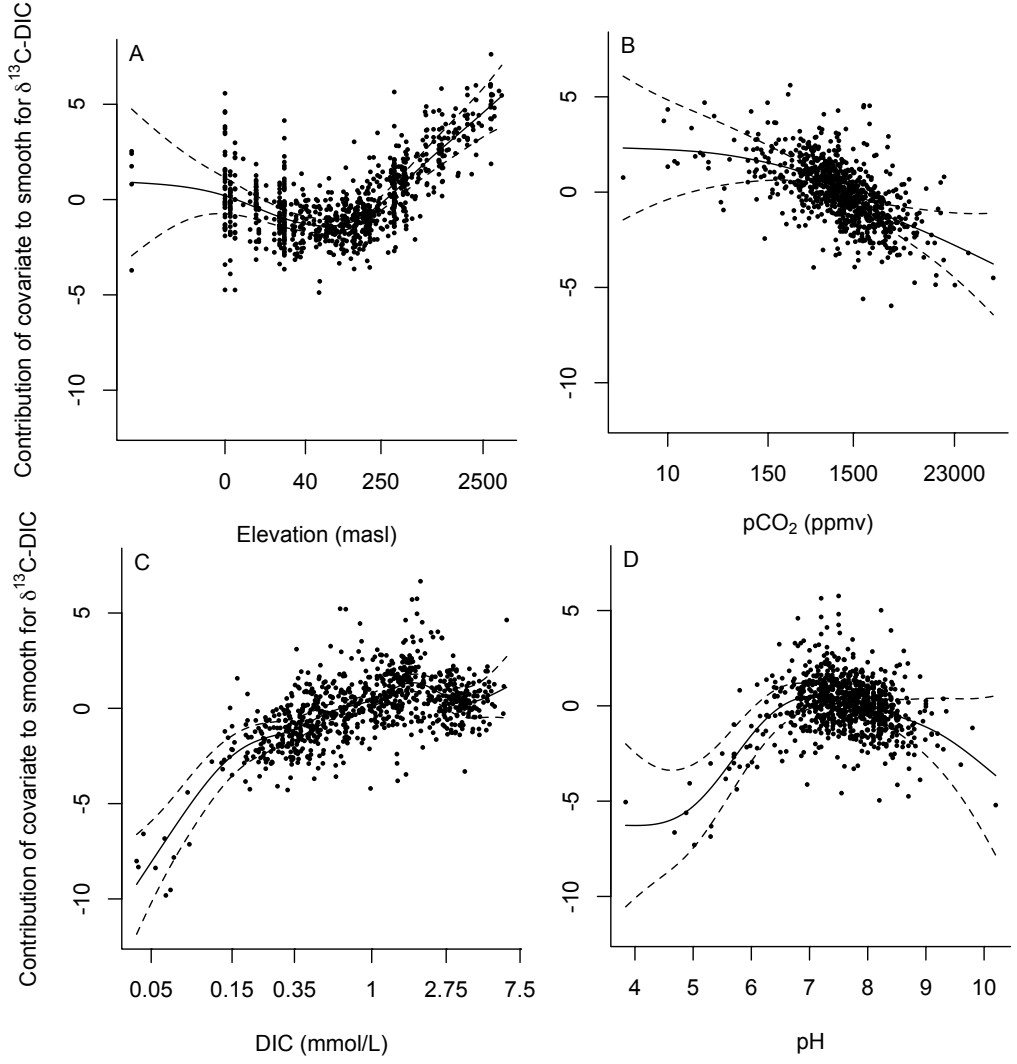


Figure 7. Smooth function (curve) derived from the generalized additive mixed model (GAMM)
fit for relationships between rescaled $\delta^{13}$C-DIC and elevation (masl, A), pCO$_2$ (ppmv, B),
dissolved inorganic carbon (DIC) concentrations (mmol/L, C), and pH (D). Dashed curves
represent 95% confidence intervals for the smooth. Partial residuals (points) are also shown.
Note that the smooth function y-axis is centered to zero mean.





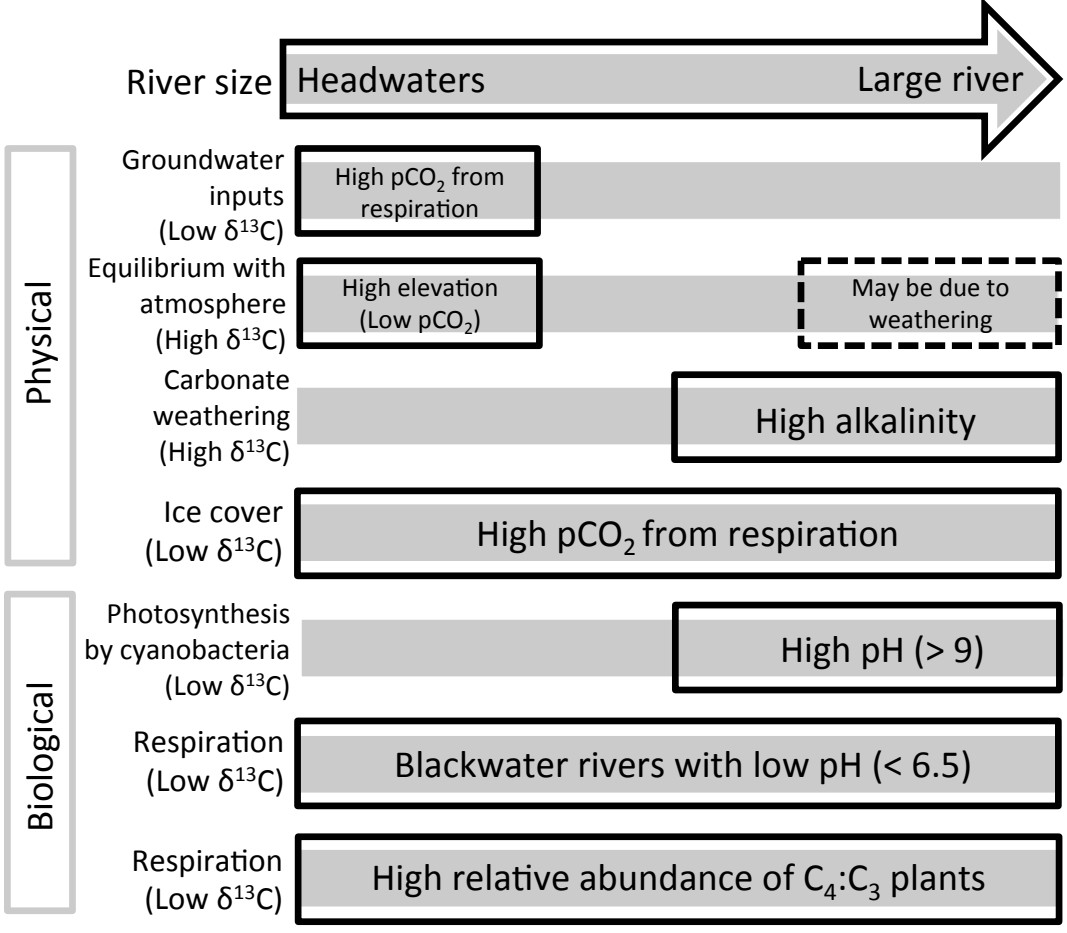



Figure 8. Schematic diagram showing dominant physical and biological controls of $\delta^{13}$C-DIC in

rivers throughout the world. The grey figures represent the longitudinal river gradient. The black

outlines indicate the relative importance of various processes influencing $\delta^{13}$C-DIC along the

upstream-downstream gradient. Patterns in $pCO_2$ along the river gradient also are shown. The

dotted outline indicates that floodplain rivers are near isotopic equilibrium with the atmosphere,

but this may be because carbonate weathering also is associated with high $\delta^{13}$C-DIC values.