# Peer review of "Controls of longitudinal variation in $\delta^{13}$ C-DIC in rivers: A"

_Biogeosciences, 2015_

## Referee Comment (RC1) · Anonymous Referee #1 · 19 Jan 2016

MAJOR COMMENTS

I suggest that the authors make their data-base public (as a supplement) for future studies and comparisons. The compilation is far from complete, the authors missed numerous papers (see below). I was not convinced from such a broad analysis that cyanobacteria have a major influence on the delta13C-DIC distribution in rivers.

SPECIFIC COMMENTS

I suggest the authors use the term "isotopic composition" instead of "isotopic signature". A signature is individual and invariable, which is not the case of stable isotopes.

L 23 : What's the difference between "decomposition" and "respiration" ?

L55 : HCO3- is also formed from weathering of silicate rocks
[Figure]

L55 : CaCO3 precipitation can also occur in rivers at travertines (Herman & Lorah 1987) or in highly eutrophic rivers (Abril et al. 2003).

L 269: replace "carbonate dissolution" by carbonate rock weathering

L 284 : statement "pCO2 in rivers tends to decline downstream" is to some extent correct in temperate rivers with limited wetlands. It is not correct in tropical rivers where the presence of floodplains and wetlands in lowlands leads to increases in pCO2 downstream (Abril et al. 2014; Borges et al. 2015a;b).

L 286 : statement "tropical rivers typically have higher concentrations of CO2 (aq) than temperate rivers" is not entirely true (anymore). Since the Aufdenkampe paper there have several studies rivers that show that pCO2 is very variable in tropical rivers (e.g. Borges et al. 2015a), and values are frequently lower than the "tropical average" given by Aufdenkampe.

L 301-303 : this statement is incorrect. The two largest rivers in the world (Amazon and Congo) have extremely low carbonate alkalinity values. The carbonate alkalinity is strongly function of latitude (that co-varies with lithology). Refer to classic work by Meybeck or Gaillardet.

L 310: This is not entirely true. While there's abundant organic carbon, black water rivers have also very low nutrient concentrations (N,P) that limit bacterial growth (Castillo et al. 2003) in addition to low pH and low O2 conditions that are not favorable to bacteria. What is important for delta 13C-DIC is that these rivers have very low HCO3- and most of the DIC is CO2.

L 336 : Abril et al. (2014) showed that root respiration by living macrophytes also contributes directly to CO2 in the water. Regarding the importance of C4 inputs in tropical watersheds refer to Marwick et al. (2014).

It could useful to try to include wetlands/floodplains into Figure 8.

There are numerous studies that were missed in the data compilation that should be

updated. A much more scrupulous search of literature is needed. Here's a list that came from the top of my mind:

Abril et al. (2013)

Balagizi et al. (2015)

Borges et al. (2014)

Brunet et al. (2009)

Darling et al. (2016)

Dubois et al. (2010)

Polsenaere & Abril (2012)

Teodoru et al. (2015)

REFERENCES

Abril et al. (2003) Carbonate dissolution in the turbid and eutrophic Loire estuary. Marine Ecology, Progress Series. 259: 129-138

Abril et al. (2013) Export of 13C-depleted dissolved inorganic carbon from a tidal forest bordering the Amazon Estuary. Estuarine Coastal and Shelf Science 129 : 23-27.

Abril, G. et al. (2014).Amazon River carbon dioxide outgassing fuelled by wetlands. Nature 505, 395-398

Balagizi et al. (2015) River geochemistry, chemical weathering and atmospheric CO2 consumption rates in the Virunga Volcanic Province (East Africa), Geochemistry, Geophysics, Geosystems (G-Cubed), 16, 2637–2660, doi:10.1002/2015GC005999

Borges et al. (2014) Carbon cycling of Lake Kivu (East Africa): net autotrophy in the epilimnion and emission of CO2 to the atmosphere sustained by geogenic inputs, PLoS ONE 9(10): e109500. doi:10.1371/journal.pone.0109500

Borges et al. (2015a) Globally significant greenhouse gas emissions from African inland waters, Nature Geoscience, 8, 637-642

Borges et al. (2015b) Divergent biophysical controls of aquatic CO2 and CH4 in the World's two largest rivers, Scientific Reports, 5:15614, doi: 10.1038/srep15614

Brunet et al. (2009) Terrestrial and fluvial carbon fluxes in a tropical watershed: Nyong basin, Cameroon, Chemical Geology, 265, 563–572

Castillo et al. (2003) Bottom-up controls on bacterial production in tropical lowland rivers. Limnol. Oceanogr. 48, 1466-1475.

Darling et al. (2016) A long-term study of stable isotopes as tracers of processes governing water flow and quality in a lowland river basin: the upper Thames, UK, Hydrological Processes, DOI: 10.1002/hyp.10779

Dubois, K. D., D. Lee, and J. Veizer (2010), Isotopic constraints on alkalinity, dissolved organic carbon, and atmospheric carbon dioxide fluxes in the Mississippi River, J. Geophys. Res., 115, G02018, doi:10.1029/2009JG001102.

Herman SH and MM Lorah (1987) CO2 outgassing and calcite precipitation in falling spring creek, Virginia, U.S.A. Chemical Geology, 62, 251-262 251

Marwick et al. (2014) Disproportionate contribution of riparian inputs to organic carbon pools in freshwater systems, Ecosystems, 17: 974–989, doi:10.1007/s10021-014-9772-6

Polsenaere & Abril (2012) Modelling CO2 degassing from small acidic rivers using water pCO2, DIC and $\delta$13C-DIC data. Geochimica et Cosmochimica Acta. 91: 220–239.

Teodoru et al. (2015) Dynamics of greenhouse gases (CO2, CH4, N2O) along the Zambezi River and major tributaries, and their importance in the riverine carbon budget, Biogeosciences, 12, 2431–2453

---

## Referee Comment (RC2) · Anonymous Referee #2 · 14 Feb 2016

General comments:

This paper summarizes current knowledge on factors determining $\delta 13C$ of DIC in stream water. As shown in many previous studies, the mechanism is highly complicated and the variables are usually inter-correlated. Furthermore, the effects are sometimes nonlinear and this seems to be why the authors adopted the GAMM. I'm not confident that this approach is valid because the mechanism lies on multi-scale (both in space and time), showing a hierarchical structure. In this viewpoint, structural equation modelling or path analysis may be better approach to deal with this type of analysis. If the authors can show a certain advantage of GAMM over a hierarchical approach, it should be explained in text. Another problem is that the authors did not quantitatively show the uncertainty in their results. I think the parameters are too many to explain the observed range in $\delta 13C$-DIC. Dissolved atmospheric CO2 and carbon-

ate bedrocks in particular show overlapping $\delta$13C values (ca. $\sim$ 0 permil) in general. Therefore, it is usually difficult to estimate relative contributions of these isotopically similar endmembers to stream water DIC. The use of another isotope (e.g., 14C) may be one of the solutions for this "too many sources problem". For example, Ishikawa et al. (2015) Radiocarbon measured $\delta$13C and $\Delta$14C of DIC and tried to estimate their sources. Although there are not many $\Delta$14C-DIC data available yet compared with $\delta$13C-DIC, $\Delta$14C-DIC may be useful for understanding potential controls of spatiotemporal variations in carbon isotopic compositions of DIC. Overall, I acknowledge the authors' effort for collecting the literature data, but the manuscript is not ready for immediate publication. Since this study is potentially important for the biogeochemical science, the authors should revise the manuscript especially focusing on my comments below.

Specific comments:

L.36: "altered" should be replaced with "determined"

L.53-54: "At isotopic $\sim$ and CO32-" Meaningless sentence so delete

L.55: "DIC(aq)" should be replaced with "DIC"

L.60: "For example $\sim$ $\delta$13C-DIC" You forget to say groundwater DIC is generally 13C-depleted

L.70: "$\delta$13C signature of DIC at isotopic equilibrium with the atmosphere" You already defined "$\delta$13C-DICequilibrium" above so call it hereafter

L.103: Can you show working hypotheses of this study at the last paragraph of the Introduction? Then explain why you focused on each of the variables and how you expected the results

Results section should be re-organized because many topics are scattered and not in order

[Figure]

L.229: "Again, $\sim$ nonlinear" This result is already reported in L. 209-211

Discussion section is relatively long so should be divided by several subsections

L.237: Start with main finding, not objective, of this study

L.254: "low surface:volume ratio" Needs more explanation. I expect high elevation (headwater?) streams are shallow in depth and narrow in width

L.261: "likely low" Remove "likely". Do you mean "near zero"?

L.266: "$\sim$ values also were low $\sim$" Do you mean high DIC concentration is due to large proportion of carbonate dissolution?

L.268-270: "$\Delta\delta$13C-DIC $\sim$ $\delta$13C signature" A gap in logic. Carbonate (e.g., limestone bedrock) has higher $\delta$13C value than atmospheric CO2. Weathering (dissolution) of carbonates provides high $\delta$13C into water column. But note that dissolved atmospheric CO2 typically shows a similar $\delta$13C value with that of carbonates

L.277-281: "Although algal $\sim$ by algae" Unclear. The second sentence does not connect well with the first sentence

L.288-289: "The cycling $\sim$ $\delta$13C-DIC" This is a principal of your analysis and should be appeared earlier in discussion

L.289 ", thus $\sim$" this statement is already mentioned just before this clause. Redundant

L.294-295: "Most lotic $\sim$ the atmosphere" This is already mentioned above

L.295: "Therefore $\sim$" Given CO2 in most streams is supersaturated, CO2 output should rise above input. You already mentioned that streams are source but not sink of CO2

L.300: "Our results also $\sim$" Redundant. Is this because of carbonate?

L.303: "buffering capacity" What is this? Unclear

P.323: The authors do not directly answer the question here: why seasonal shift in

$\delta$13C-DIC in high latitude is observed?

L.336: "Mayorga et al. (2005)" Another important contribution of this paper was that they measured radiocarbon ($\Delta$14C) of DIC as well as other organic carbon fractions. I strongly recommend the authors also mention $\Delta$14C because it can separate sources (e.g., dissolved atmospheric CO2 and carbonate bedrocks) that cannot be separated by $\delta$13C. See also Raymond et al. (2004) Marine Chemistry and references therein

L.336-339: "Terrestrial C4 $\sim$ low-water periods" But C4 plants have higher $\delta$13C values than C3, don't they?

L.337: "terrestrial C3" Remove "terrestrial". Redundant

Fig. 3: Are panels A and B same? They look very similar

Figs. 4, 7, and their legends: Please explain how you calculated y axis (Contribution of covariate to smooth for $\Delta\delta$13C-DIC or $\delta$13C-DIC)

Fig. 6: Seasonal pattern seems to be different between northern and southern hemispheres. Why? Just because of number of data?

———————————————————

---

## Author Comment (AC1) · 9 Mar 2016

Dear Anonymous Reviewer 1,

We have made extensive revisions to our manuscript "Controls of longitudinal variation in $\delta$13C-DIC in rivers: A global meta-analysis." The excellent suggestions allowed us to critically review the manuscript, and as a result the paper is much improved. We have increased the number of studies from our meta-analysis from 26 to 31. We also changed one of the covariates in our GAMMs from DIC concentration to bicarbonate (HCO3-) concentration. We did this because we originally used DIC as a proxy for weathering, following Bade et al. (2004), however we realized that substituting DIC for HCO3- would increase the number of data points included in the GAMMs from 889 to 2,087, resulting in much better geographic coverage. In addition, we appreciate the

suggestion to provide our data-base as a supplement for future studies. Below are detailed comments justifying how the manuscript was revised. We are ready to submit our revised manuscript and data-base upon request by the editor. We hope that you will find the revised manuscript suitable for publication in Biogeosciences.

Sincerely, Katherine A. Roach on behalf of the authors

Reviewer comment: I suggest that the authors make their data-base public (as a supplement) for future studies and comparisons. The compilation is far from complete, the authors missed numerous papers (see below). I was not convinced from such a broad analysis that cyanobacteria have a major influence on the delta13C-DIC distribution in rivers.

Author comment: We will provide our dataset to Biogeosciences as supplementary material so other researchers can use it in the future. As explained below, we could not include all of the studies that this reviewer recommended. In the manuscript, we clarified that we constrained the $\delta$13C-DIC data to only include samples that measured temperature, pH, and one of the following variables that could be used to assess the relative concentrations of $H_2CO_3$, $HCO_3-$, and $CO_3^{2-}$: DIC concentrations, alkalinity, or $HCO_3-$ concentrations in rivers with low to neutral pH. We agree the assertion that cyanobacteria have a major influence on $\delta$13C-DIC in rivers is speculative. However, intense photosynthesis by cyanobacteria has been shown to produce fractionation that results in low $\delta$13C-DIC values when pH is high and pCO2 is low (Herczog and Fairbanks, 1987), and our results indicated that $\delta$13C-DIC values were low in rivers with high pH and low pCO2 values. Following the addition of new data, sites with the highest pH no longer tend to have low $\delta$13C-DIC values, therefore we removed the discussion of cyanobacteria from the Discussion section.

Herczog, A.L., and Fairbanks, R.G. Anomalous carbon isotope fractionation between atmospheric CO2 and dissolved inorganic carbon induced by intense photosynthesis, Geochim. Cosmochim. Acta, 51, 895–899, 1987.

Reviewer comment: I suggest the authors use the term "isotopic composition" instead of "isotopic signature". A signature is individual and invariable, which is not the case of stable isotopes.

Author comment: As a result of this comment, we realized that the term "stable isotope signature" is the terminology that is most commonly used by biologists, whereas "stable isotope composition" seems to be the terminology that is most commonly used by biogeochemists (e.g., Mayorga et al., 2005). Our paper focuses on dissolved inorganic carbon, therefore we revised the terminology "$\delta$13C signature" to "$\delta$13C composition" or "$\delta$13C values."

Reviewer comment: L 23: What's the difference between "decomposition" and "respiration"?

Author comment: By decomposition, we meant methanogenesis followed by methane oxidation, which would produce $CO_2$ that is highly depleted in 13C. However because methane oxidation by methanotrophs is a form of respiration, we changed the terminology "decomposition" to "respiration" throughout the manuscript. Furthermore, we added the following sentences to the Introduction section to make more explicit why DIC from heterotrophic respiration is depleted in 13C: "The $\delta$13C of the $CO_2$ that is produced during heterotrophic respiration is similar to the $\delta$13C of the energy source. In rivers the energy source is often plants, which are relatively depleted in 13C (Deines, 1980; Kohn, 2010). Furthermore, methanogenesis in anaerobic sediments produces methane that is extremely depleted in 13C, and this methane can then be oxidized to $CO_2$ by methanotrophs (Conrad et al., 2011)."

Conrad, R., Noll, M., Claus, P., Klose, M., Bastos, W.R., and Enrich-Prast, A. Stable carbon isotope discrimination and microbiology of methane formation in tropical anoxic lake sediments, Biogeosciences, 8, 795–814, 2011.

Deines, P. The isotopic composition of reduced organic carbon, in: Handbook of Environmental Isotope Geochemistry, Vol. 1, edited by: Fritz, P., and Fontes, J.C., New

York, Elsevier, 329–406, 1980.

Kohn, M.J. Carbon isotope composition of terrestrial C3 plants as indicators of (paleo)ecology and (paleo)climate, Proc. Natl. Acad. Sci. USA, 16, 19691–19695, 2010.

Reviewer comment: L 55: HCO3- is also formed from weathering of silicate rocks.

Author comment: We changed the sentence "Other physical processes influencing $\delta$13C-DIC in rivers include carbonate mineral weathering and mixing of different water bodies" to "Other physical processes influencing $\delta$13C-DIC in rivers include chemical weathering of rocks and mixing of different water bodies." We also explained that "The major source of DIC in most rivers is carbonate weathering, the formation of HCO3- via the dissolution of carbonate minerals (Raymond et al., 2008)."

Reviewer comment: L 55: CaCO3 precipitation can also occur in rivers at travertines (Hernan and Lorah 1987) or in highly eutrophic rivers (Abril et al. 2003).

Author comment: It would have been interesting to include Ca2+ as a covariate in the GAMMs to investigate if carbonate precipitation influences $\delta$13C-DIC in rivers. Unfortunately, we only have data on Ca2+ in eighteen of the thirty-one studies in our meta-analysis. Most of the world's major river systems are not supersaturated in Ca2+ and HCO3- (Kempe 1982) thus carbonate precipitation at travertines likely does not have a large effect on $\delta$13C-DIC in most fluvial ecosystems."

We also were interested in investigating if eutrophication had an effect on $\delta$13C-DIC in rivers. However, variables that are associated with eutrophication, such as chlorophyll a or algal primary production, were rarely measured. We added the following sentence to the Discussion section: "Algal primary production or other processes that are associated with eutrophication might account for some of the unexplained variation in the GAMM analyses of $\Delta\delta$13C-DIC and $\delta$13C-DIC. For example, calcium carbonate precipitation can occur in highly eutrophic rivers (Abril et al., 2003), and this process causes fractionation of DIC (Emrich et al., 1970)."

Emrich, K., Ehhalt, D.H., and Vogel, J.C. Carbon isotope fractionation during the precipitation of calcium carbonate, Earth Planet. Sci. Lett., 8, 363–371, 1970. Kempe S. 1982. Long-term records of $CO_2$ pressure fluctuations in fresh waters. SCOPE/UNEP Sonderband 52:91-332.

Reviewer comment: L 269: replace "carbonate dissolution" by carbonate rock weathering.

Author comment: We replaced "carbonate dissolution" with "carbonate rock weathering."

Reviewer comment: L 284: statement "pCO2 in rivers tends to decline downstream" is to some extent correct in temperate rivers with limited wetlands. It is not correct in tropical rivers where the presence of floodplains and wetlands in lowlands leads to increases in pCO2 downstream (Abril et al. 2014; Borges et al. 2015a;b).

Author comment: Originally, most of the data in our meta-analysis were from temperate rivers, which is why we stated that pCO2 in these rivers tends to decline downstream. The additional data we added to the meta-analysis were almost all from tropical rivers. Furthermore, with the addition of the new data, pCO2 is no longer included in the GAMM of $\delta$13C-DIC. Therefore, we removed this statement during manuscript revision.

Reviewer comment: L 286: statement "tropical rivers typically have higher concentrations of CO2 (aq) than temperate rivers" is not entirely true (anymore). Since the Aufdenkampe paper several studies have shown that pCO2 is very variable in tropical rivers (e.g., Borges et al. 2015a), and values are frequently lower than the "tropical average" given by Aufdenkampe.

Author comment: We appreciate this new information on global patterns in CO2 (aq) concentrations in rivers. With the addition of the new data, pCO2 is no longer included in the GAMM of $\delta$13C-DIC. Therefore, we removed this statement during manuscript revision.

Reviewer comment: L 301-303: this statement is incorrect. The two largest rivers in the world (Amazon and Congo) have extremely low carbonate alkalinity values. The carbonate alkalinity is strongly a function of latitude (that co-varies with lithology). Refer to classic work by Meybeck or Gaillardet.

Author comment: Originally, most of the data in our meta-analysis were from temperate rivers, in which carbonate alkalinity has been documented to increase with increasing river size (e.g., Telmer and Veizer 1999). The additional data we added to the meta-analysis were almost all from tropical rivers. Furthermore, with the addition of the new data, Strahler order is no longer included in the GAMM of $\delta$13C-DIC. Therefore, we removed this statement during manuscript revision. In addition, we clarified that "Most tropical rivers are highly weathered and thus their waters tend to be more dilute in dissolved materials than temperate rivers (Gaillardet, 1997)."

Gaillerdet, J., Dupré, B., Allègre, C.J., and Négel, P. Chemical and physical denudation in the Amazon River Basin, Chem. Geol., 142, 141–173, 1997.

Telmer, K.H., and Veizer, J. Carbon fluxes, pCO2 and substrate weathering in a large northern river basin, Canada: carbon isotope perspectives, Chem. Geol., 159, 61–86, 1999.

Reviewer comment: L 310: This is not entirely true. While there's abundant organic carbon, black water rivers also have very low nutrient concentrations (N,P) that limit bacterial growth (Castillo et al. 2003) in addition to low pH and low O2 conditions that are not favorable to bacteria. What is important for delta 13C-DIC is that these rivers have very low HCO3- and most of the DIC is CO2.

Author comment: We revised this sentence and explained that "In blackwater rivers, alkalinity is extremely low and the DIC pool is mainly composed of biogenic CO2 derived from respiration in organic sediments and soils which has a low $\delta$13C signature (Rau, 1978; Tan and Edmond, 1993)."

Reviewer comment: L 336: Abril et al. (2014) showed that root respiration by living macrophytes also contributes directly to CO2 in the water. Regarding the importance of C4 inputs in tropical watersheds refer to Marwick et al. (2014).

Author comment: We added a sentence explaining that "The period of floodplain inundation is associated with heterotrophic respiration of organic matter and root respiration by living macrophytes (Richey et al., 1988; Langhans and Tockner, 2006; Abril et al., 2014) that lowers $\Delta\delta$13C-DIC and $\delta$13C-DIC."

We revised the manuscript to focus on the importance of respiration of organic matter of terrestrial origin (rather than focusing on C4 grasses and C3 plants separately) to $\delta$13C-DIC in tropical rivers. In doing so, we added a sentence stating that "Studies in tropical rivers have found that terrestrial plants contribute a large amount of material to both the inorganic and organic carbon pools (Mayorga et al., 2005; Marwick et al., 2014)."

Abril, G., Martinez, J.M., Artigas, L.F., Moreira-Turcq, P., Bernedetti, M.F., Vidal, L., Meziane, T., Kim, J.H., Bernardes, M.C., Savoye, N., Deborde, J., Souza, E.L., Albéric, P., Landim de Souza, M.F., and Roland, F. Amazon River carbon dioxide outgassing fuelled by wetlands, Nature, 505, 395–398, 2014.

Langhans, S.D., and Tockner, K. The role of timing, duration, and frequency of inundation in controlling leaf litter decomposition in a river-floodplain ecosystem (Tagliamento, northeastern Italy), Oecologia, 147, 501–9, 2006.

Marwick, T.R., Borges, A.V., Van Acker, K., Darchambeau, F., and Bouillon, S. Disproportionate contribution of riparian inputs to organic carbon pools in freshwater systems, Ecosystems, doi:1007/s10021-014-9772-6, 2014.

Richey, J.E., Devol, A.H., Wofsy, S.C., Victoria, R., and Ribeiro, M.N.G. Biogenic gases and the oxidation and reduction of carbon in the Amazon River and floodplain waters, Limnol. Oceanogr., 33, 551–561, 1988.

Reviewer comment: It could be useful to try to include wetland/floodplains into Figure 8.

Author comment: We also thought it would be interesting to classify each of the sites to reflect whether they were located on a floodplain. We previously considered using the following global inundation map to classify our sites using the mean annual maximum land surface inundation extent:

Fluet-Chouinard, E., Lehner, B., Rebelo, L.M., Papa, F., and Hamilton, S.K. Development of a global inundation map at high spatial resolution from topographic downscaling of coarse-scale remote sensing data, Remote Sens. Environ., 158, 348–361, 2015.

We contacted the first author for directions on how to download the map and she told us "A word of warning: GIEMS-D15 isn't accurate on the local scale and include various other types of inundated areas than simply floodplains. Depending on the geographical extent of your sampling sites it'll be important not to over-interpret the presence of GIEMS-D15-MAMax over your sites. I am currently working on a classified version of GIEMS-D15 that will separate major wetland categories, but that is still very much in the works."

We decided against classifying our sites using a map that is not accurate at the local scale and includes areas other than floodplains.

Reviewer comment: There are numerous studies that were missed in the data compilation that should be updated. A much more scrupulous search of literature is needed. Here's a list that came from the top of my mind: Abril et al. (2013), Balagizi et al. (2015), Borges et al. (2014), Brunet et al. (2009), Darling et al. (2016), Dubois et al. (2010), Polsenaere and Abril (2012), Teodoru et al. (2015)

Author comment: We appreciate the suggestion to add these papers to our meta-analysis. As indicated in the Methods section, we collated data from the scientific

literature using the search engines Google Scholar and Web of Science and using the search terms "river", "stable isotope", and "dissolved inorganic carbon". We did not find any of these papers during our search, however we should have been more explicit about the dates that were searched. We included the following sentence in the Methods section to be more explicit: "Papers included in our meta-analysis were published prior to the year 2016." We included Balagizi et al. (2015), Brunet et al. (2009), Dubois et al. (2010), and Teodoru et al. (2016) in our revised meta-analysis. We did not include Abril et al. (2013) or Borges et al. (2014) because Abril et al. (2013) sampled the Amazon estuary and Borges et al. (2014) sampled Lake Kivu, a large lake in East Africa, whereas our meta-analysis focused on rivers. Darling et al. (2016) is not yet published in the journal Hydrological Processes. According to the journal, the final edited and typeset version of record will appear in the future. Polsenare and Abril (2012) developed a model to predict $CO_2$ degassing in small headwater streams using $pCO_2$, DIC, and $\delta$13C-DIC. The original data used to develop the model was published in: Polsenare, P., N. Savoye, H. Etcheber, M. Canton, D. Poirier, S. Bouillon, and G. Abril. 2013. Export and degassing of terrestrial carbon through watercourses draining a temperate podsolised catchment. Aquatic Sciences 75:299-319. Polsenare et al. (2013) did not provide information on pH, a key variable in our meta-analysis.

---

## Author Comment (AC2) · 9 Mar 2016

Dear Anonymous Reviewer 2,

We have made extensive revisions to our manuscript "Controls of longitudinal variation in $\delta$13C-DIC in rivers: A global meta-analysis." The excellent suggestions allowed us to critically review the manuscript, and as a result the paper is much improved. We have increased the number of studies from our meta-analysis from 26 to 31. We also changed one of the covariates in our GAMMs from DIC concentration to bicarbonate (HCO3-) concentration. We did this because we originally used DIC as a proxy for weathering, following Bade et al. (2004), however we realized that substituting DIC for HCO3- would increase the number of data points included in the GAMMs from 889 to 2,087, resulting in much better geographic coverage. We particularly appreciate the

suggestions on the paper's organization, including adding working hypotheses in the Introduction section and dividing the Discussion sections into sub-sections. Below are detailed comments justifying how the manuscript was revised. We are ready to submit our revised manuscript and data-base upon request by the editor. We hope that you will find the revised manuscript suitable for publication in Biogeosciences.

Sincerely, Katherine A. Roach on behalf of the authors

Reviewer comment: This paper summarizes current knowledge on factors determining $\delta$13C of DIC in stream water. As shown in many previous studies, the mechanism is highly complicated and the variables are usually inter-correlated. Furthermore, the effects are sometimes nonlinear and this seems to be why the authors adopted the GAMM. I'm not confident that this approach is valid because the mechanism lies on multi-scale (both in space and time), showing a hierarchical structure. In this viewpoint, structural equation modelling or path analysis may be better approach to deal with this type of analysis. If the authors can show a certain advantage of GAMM over a hierarchical approach, it should be explained in text. Another problem is that the authors did not quantitatively show the uncertainty in their results. I think the parameters are too many to explain the observed range in $\delta$13C-DIC. Dissolved atmospheric CO2 and carbonate bedrocks in particular show overlapping $\delta$13C values (ca. âĹij 0 permil) in general. Therefore, it is usually difficult to estimate relative contributions of these isotopically similar endmembers to stream water DIC. The use of another isotope (e.g., 14C) may be one of the solutions for this "too many sources problem". For example, Ishikawa et al. (2015) Radiocarbon measured $\delta$13C and $\Delta$14C of DIC and tried to estimate their sources. Although there are not many $\Delta$14C-DIC data available yet compared with $\delta$13C-DIC, $\Delta$14C-DIC may be useful for understanding potential controls of spatiotemporal variations in carbon isotopic compositions of DIC. Overall, I acknowledge the authors' effort for collecting the literature data, but the manuscript is not ready for immediate publication. Since this study is potentially important for the biogeochemical science, the authors should revise the manuscript especially focusing

on my comments below.

Author comment: Our GAMMs are multi-level hierarchical models, with three levels of spatial structure (site, watershed, and river). We did not use a temporally hierarchical model because the data from the literature survey did not allow for this type of analysis. We added an additional sentence in the Methods section of the manuscript explaining that our GAMMs are hierarchical models that represent an improvement over previous analyses of $\delta$13C-DIC using correlations and multiple regressions (e.g., Bade et al., 2004) because they account for the multiscale structure (sites, rivers, watersheds) of the data and allow for nonlinear relationships.

The uncertainty in the GAMM fits is reflected in the 95% confidence intervals for the smooths that are shown by the dashed lines in Figures 4 and 6. This is explained in the figure legends. As we explain in the Methods section, for each of the GAMMs, we iteratively removed variables if the 95% confidence intervals for the smooth function included zero throughout the range of measured values. Additionally, the p values provided in Tables 1 and 2 provide an indication of the usefulness of individual predictors in the GAMMs.

We realize that dissolved atmospheric CO2 and carbonate bedrock both have high $\delta$13C-DIC values. We attempted to make this clearer by adding an additional hypothesis in the Introduction section explaining that "We anticipated that $\Delta\delta$13C-DIC and $\delta$13C-DIC values would increase with HCO3- concentration, consistent with carbon derived from carbonate rock weathering." In addition, we added the following sentence to the Discussion section: "Because dissolved atmospheric CO2 and DIC derived from carbonate rock weathering have similar $\delta$13C values, the use of $\Delta$14C as an additional tracer would result in more effective differentiation between these sources (e.g., Raymond et al., 2004; Ishikawa et al., 2015)." We agree that it would have been interesting to include $\Delta$14C-DIC as a covariate in the GAMMs, however this variable was only measured for four of the thirty-one studies from our meta-analysis.

[Figure]

Ishikawa, N.F., Tayasu, I., Yamane, M., Yokoyama, Y., Sakai, S., and Ohkouchi, N. Sources of dissolved inorganic carbon in two small streams with different bedrock geology: insights from carbon isotopes, Radiocarbon, 57, 439–448, 2015.

Raymond, P.A., Bauer, J.E., Caraco, N.F., Cole, J.J., Longworth, B., and Petsch, S.T. Controls on the variability of organic matter and dissolved inorganic carbon ages in northeast US rivers, Mar. Chem., 92, 353–366, 2004.

Reviewer comment: L.36: "altered" should be replaced with "determined"

Author comment: We changed the sentence "The $\delta$13C signature of DIC ($\delta$13C-DIC) in the water column can be altered by the addition of DIC with a distinctive $\delta$13C signature and by processes that affect the relative abundance of 13C:12C (fractionation)" to "The $\delta$13C signature of DIC ($\delta$13C-DIC) in the water column can be modified by the addition of DIC with a distinctive $\delta$13C value and by processes that affect the relative abundance of 13C:12C (fractionation)." We use "modify" here in the conventional sense: to change somewhat the form of qualities of; alter partially; amend.

Reviewer comment: L.53-54: "At isotopic and CO32-" Meaningless sentence so delete

Author comment: We don't agree that the sentence "At isotopic equilibrium with the atmosphere, CO2 (aq) has a lower $\delta$13C signature relative to HCO3- and CO32-" is meaningless. We simply meant that when pH is low and the DIC pool is dominated by CO2, DIC at isotopic equilibrium with the atmosphere has a lower $\delta$13C signature than when pH is high and the DIC pool is dominated by HCO3- or CO32-. However we attempted to clarify this sentence by revising it to "$\delta$13C-DIC values at isotopic equilibrium with the atmosphere ($\delta$13C-DICequilibrium) are low in streams with acidic surface water (dominant in CO2; approximately -10‰ to -2‰ relative to values in streams with neutral to basic pH (dominant in HCO3-; approximately -2‰ to 2‰."

Reviewer comment: L.55: "DIC(aq)" should be replaced with "DIC"

Author comment: We replaced DIC(aq) with DIC.

[Figure]

Reviewer comment: L.60: "For example $\delta$13C-DIC" You forget to say groundwater DIC is generally 13Cdepleted

Author comment: We revised the sentence "Groundwater is supersaturated in CO2 (aq) from soil respiration and decomposition of organic matter, and its input lowers $\delta$13C-DIC" to "Groundwater has DIC that is depleted in 13C (low $\delta$13C values) from soil respiration and its input lowers instream $\delta$13C-DIC."

Reviewer comment: L.70: "$\delta$13C signature of DIC at isotopic equilibrium with the atmosphere" You already defined "$\delta$13C-DICequilibrium" above so call it hereafter

Author comment: We moved the definition $\delta$13C-DICequilibrium to the second paragraph of the Introduction section.

Reviewer comment: L.103: Can you show working hypotheses of this study at the last paragraph of the Introduction? Then explain why you focused on each of the variables and how you expected the results

Author comment: We revised the last paragraph of the Introduction section and now explain our expectations for relationships between $\Delta\delta$13C-DIC and/or $\delta$13C-DIC and pCO2, elevation, HCO3- concentrations, and Strahler order. We explained that "We originally expected that if $\delta$13C-DIC was mainly under biotic control, we would find a negative relationship between pCO2 and $\Delta\delta$13C-DIC. We expected to find low $\Delta\delta$13C-DIC values in high-elevation streams because of significant CO2 outgassing in these systems. We anticipated that $\Delta\delta$13C-DIC and $\delta$13C-DIC values would increase with HCO3- concentration, consistent with carbon derived from carbonate rock weathering. Finally, we expected that $\delta$13C-DIC values would be positively related to river size, as measured by Strahler order, because of a decrease in groundwater inputs and an increase in CO2 loss to the atmosphere and algal primary production with increasing river size."

Reviewer comment: Results section should be re-organized because many topics are

scattered and not in order

Author comment: We believe the Results section is ordered logically. We first provide information on the full range in observed values of $\delta$13C-DIC and $\delta$13C-DICequilibrium. We then discuss results from the GAMM of $\Delta\delta$13C-DIC, including the explanatory variables retained, the deviance explained, and relationship between $\Delta\delta$13C-DIC and each of the explanatory variables. We then discuss results from the GAMM of $\delta$13C-DIC, including the explanatory variables retained, the deviance explained, and the relationships between $\delta$13C-DIC and each of the explanatory variables.

Reviewer comment: L.229: "Again, nonlinear" This result is already reported

Author comment: This paragraph explains GAMM results for $\delta$13C-DIC, and the previous paragraphs explains GAMM results for $\Delta\delta$13C-DIC (deviations between $\delta$13C-DIC and $\delta$13C-DICequilibrium).

Reviewer comment: L. 209-211 Discussion section is relatively long so should be divided by several subsections

Author comment: We have divided the Discussion section into three subsections: "Processes influencing $\Delta\delta$13C-DIC and $\delta$13C-DIC", "Seasonal shifts in $\Delta\delta$13C-DIC and $\delta$13C-DIC", and "Longitudinal shifts in processes controlling $\delta$13C-DIC."

Reviewer comment: L.237: Start with main finding, not objective, of this study

Author comment: We revised the first sentence of the Discussion section from "Our main objective was to investigate controls and spatial and temporal patterns of $\delta$13C-DIC in rivers throughout the world" to "Overall, our analysis indicates that processes that add DIC to the water column such as respiration of terrestrial organic matter have a greater influence on $\delta$13C-DIC than processes that remove DIC."

Reviewer comment: L.254: "low surface:volume ratio" Needs more explanation. I expect high elevation (headwater?) streams are shallow in depth and narrow in width

Author comment: We changed "the high gradient and low surface:volume ratio in these ecosystems increases water turbulence and promotes CO2 outgassing" to "the high gradient and shallow depth of these ecosystems increases water turbulence and promotes CO2 outgassing."

Reviewer comment: L.261: "likely low" Remove "likely". Do you mean "near zero"?

L.266: "values also were low" Do you mean high DIC concentration is due to large proportion of carbonate dissolution?

Author comment: We revised these minor issues during manuscript revision.

Reviewer comment: L.268-270: "$\Delta\delta$13C-DIC $\delta$13C signature" A gap in logic. Carbonate (e.g., limestone bedrock) has higher $\delta$13C value than atmospheric CO2. Weathering (dissolution) of carbonates provides high $\delta$13C into water column. But note that dissolved atmospheric CO2 typically shows a similar $\delta$13C value with that of carbonates

Author comment: We changed this sentence during manuscript revision. In addition, we added the following sentence addressing this issue to the Discussion section: "Because dissolved atmospheric CO2 and DIC derived from carbonate rock weathering have similar $\delta$13C values, the use of $\Delta$14C as an additional tracer would result in more effective differentiation between these two DIC sources (e.g., Raymond et al., 2004; Ishikawa et al., 2015)."

Reviewer comment: L.277-281: "Although algal by algae" Unclear. The second sentence does not connect well with the first sentence

Author comment: We revised these two sentences to increase clarity.

Reviewer comment: L.288-289: "The cycling $\delta$13C-DIC" This is a principal of your analysis and should be appeared earlier in discussion

L.289 ", thus" this statement is already mentioned just before this clause. Redundant

L.294-295: "Most lotic the atmosphere" This is already mentioned above

L.295: "Therefore" Given $CO_2$ in most streams is supersaturated, $CO_2$ output should rise above input. You already mentioned that streams are source but not sink of $CO_2$

L.300: "Our results also" Redundant. Is this because of carbonate?

L.303: "buffering capacity" What is this? Unclear

Author comment: We revised these minor issues.

Reviewer comment: P.323: The authors do not directly answer the question here: why seasonal shift in $\delta$13C-DIC in high latitude is observed?

Author comment: In the Discussion section we summarized what probably caused the seasonal shifts in $\Delta\delta$13C-DIC and $\delta$13C-DIC in rivers in temperate regions with seasonal snow cover: "Ice cover also has been documented to increase pCO2 in the water column of rivers, and should be responsible for the seasonal shifts in $\Delta\delta$13C-DIC and $\delta$13C-DIC values in rivers at high latitudes in the northern hemisphere."

Reviewer comment: L.336: "Mayorga et al. (2005)" Another important contribution of this paper was that they measured radiocarbon ($\Delta$14C) of DIC as well as other organic carbon fractions. I strongly recommend the authors also mention $\Delta$14C because it can separate sources (e.g., dissolved atmospheric $CO_2$ and carbonate bedrocks) that cannot be separated by $\delta$13C. See also Raymond et al. (2004) Marine Chemistry and references therein

Author comment: In the Discussion section we will stated that "Mayorga et al. (2005) analyzed $\delta$13C and $\Delta$14C (radiocarbon) of DIC, dissolved organic carbon, and multiple particulate organic carbon fractions in Amazonian rivers and concluded that high pCO2 was sustained by in situ respiration of terrestrial plants." In addition we added the following sentence in the Discussion section: "Because dissolved atmospheric $CO_2$ and DIC derived from carbonate rock weathering have similar $\delta$13C values, the use of $\Delta$14C as an additional tracer would result in more effective differentiation between these two

DIC sources (e.g., Raymond et al., 2004; Ishikawa et al., 2015)."

Reviewer comment: L.336-339: "Terrestrial C4 low-water periods" But C4 plants have higher $\delta$13C values than C3, don't they?

Author comment: We clarified our reasoning for low $\delta$13C-DIC values in tropical rivers by stating that "Most tropical rivers are highly weathered and thus their waters tend to be more dilute in dissolved materials than temperate rivers (Gaillardet, 1997). Whereas $\delta$13C-DIC in temperate rivers may be more strongly influenced by carbonate weathering, $\delta$13C-DIC in tropical rivers may be more influenced by respiration of organic matter of terrestrial origin."

Reviewer comment: L.337: "terrestrial C3" Remove "terrestrial". Redundant

Author comment: We revised the manuscript to focus on the importance of respiration of organic matter of terrestrial origin (rather than focusing on C4 grasses and C3 plants separately) to $\delta$13C-DIC in tropical rivers.

Reviewer comment: Fig. 3: Are panels A and B same? They look very similar

Author comment: As explained in the figure legend, Figure 3A shows the relationship between fitted values and $\Delta\delta$13C-DIC and Figure 3B shows the relationship between fitted values and $\delta$13C-DIC.

Reviewer comment: Figs. 4, 7, and their legends: Please explain how you calculated y axis (Contribution of covariate to smooth for $\Delta\delta$13C-DIC or $\delta$13C-DIC)

Author comment: The y axis represents the additive contribution of each covariate to the value of the dependent variable. It is the contribution made to the fitted value for that smooth function at a given value of the covariate. We added a sentence to the figure legends of Figures 4 and 6 explaining that "The y-axis represents the additive contribution of each covariate to the value of $\Delta\delta$13C-DIC" (Figure 4) and "The y-axis represents the additive contribution of each covariate to the value of $\delta$13C-DIC" (Figure 6).

Reviewer comment: Fig. 6: Seasonal pattern seems to be different between northern and southern hemispheres. Why? Just because of number of data?

Author comment: In the Discussion section we clarified why we believe there were different seasonal patterns in $\Delta\delta$13C-DIC and $\delta$13C-DIC between rivers at high latitudes and rivers at low latitudes by revising "Sample size was low in temperate regions of the southern hemisphere. A greater number of sites sampled from these rivers may result in a seasonal trend that is similar to the pattern observed in the northern hemisphere" to "In the southern hemisphere between $40°$ and $60°$, data were sparse in space and time (Fig. 5). More complete sampling from these rivers may result in a seasonal pattern similar to the pattern observed in the northern hemisphere."

––––––––––––––––––––––––––––––